# Recent Advances of Degradation Technologies Based on PROTAC Mechanism

**DOI:** 10.3390/biom12091257

**Published:** 2022-09-07

**Authors:** Mingchao Xiao, Jiaojiao Zhao, Qiang Wang, Jia Liu, Leina Ma

**Affiliations:** 1Cancer Institute of The Affiliated Hospital, Qingdao University, Qingdao 266071, China; 2School of Basic Medicine, Qingdao University, Qingdao 266071, China; 3Oncology Department, Shandong Second Provincial General Hospital, Jinan 250022, China; 4Department of Pharmacology, School of Pharmacy, Qingdao University, Qingdao 266071, China

**Keywords:** proteolysis-targeting chimeras, protein degradation technology, E3 ligase, protein of interest

## Abstract

PROTAC (proteolysis-targeting chimeras), which selectively degrades target proteins, has become the most popular technology for drug development in recent years. Here, we introduce the history of PROTAC, and summarize the recent advances in novel types of degradation technologies based on the PROTAC mechanism, including TF-PROTAC, Light-controllable PROTAC, PhosphoTAC, LYTAC, AUTAC, ATTEC, CMA, RNA-PROTAC and RIBOTACs. In addition, the clinical progress, current challenges and future prospects of degradation technologies based on PROTAC mechanism are discussed.

## 1. Introduction

PROTAC (proteolysis-targeting chimeras) is a technology that utilizes the ubiquitin-proteasome system (UPS) in cells to specifically degrade target proteins [1]. The structure of PROTAC consists of three parts: a target protein ligand, an E3 ligase ligand and a linker that connects the two [2,3]. These three parts perform, respectively, the following functions: to selectively bind to the protein of interest (POI), to recruit the E3 ubiquitin ligase in the cell, and, to “close up” the spatial “gap” between the target protein and the E3 ubiquitin ligase and form the POI-PROTAC-E3 ternary complex [4,5]. Consequently, POI is ubiquitinated by E3 ubiquitin ligase then finally recognized and degraded by the 26S proteasome [6]. 

Different from most small molecule inhibitors, antagonists and blocker drugs traditionally used in clinical, PROTAC has unique advantages [7]. First, different from traditional drugs, PROTAC does not need to act directly on the active site. When it binds to the target protein, it can induce the degradation of the target protein. Therefore, PROTACs can break through targets that are defined by traditional medicine as “undruggable” due to the lack of a significant binding pocket in the active site. Second, different from traditional drugs that need to maintain a high concentration to saturate and continuously inhibit the target, PROTAC can dissociate from the complex and re-catalyze the next cycle of degradation after inducing the degradation of the target, so the required drug concentration is low. Third, since PROTAC has a degradation effect on the target and the degradation effect does not depend on the active site of the target, it can effectively overcome the drug resistance caused by the overexpression of the target and the mutation of the active site of the target [8].

Here, we start with PROTAC technology, the earliest and fastest developing protein degradation technology (PDT), then review the progress of degradation technologies based on PROTAC mechanism in the past 20 years. Last but not the least, we systematically compared the advantages and disadvantages of various degradation technologies, and put forward many innovative and interesting ideas for the future development of PROTAC, which is the novelty of this review.

## 2. Development History of PROTAC

In 2001, the concept of “PROTAC” was first proposed in a paper published by Professor Crews and Professor Deshaies. To prove the concept, they synthesized a chimeric compound, termed protac-1, which was the first PROTAC molecule [3]. PROTAC-1 could “recruit” a special E3 ligase complex of SCF^β-TRCP^ to the target protein MetAp-2, and then degrade MetAp-2 through the endogenous ubiquitin-proteasome degradation pathway. Following the publication of that landmark article, similar PROTAC molecules were gradually developed, such as PROTAcs that can specifically degrade androgen receptors (AR) and estrogen receptors (ER) [9,10,11]. However, the structures of all these PROTACs contain a small peptide, so both the cell permeability and cell viability are not optimal, which serves as a major limitation for the development of PROTACs. Those PROTACs were called first-generation PROTACs in later years. In 2008, a PROTAC based entirely on small molecules appeared. In that newly developed PROTAC, one end of the molecule was able to act as a ligand that selectively binds to the AR, and the other end of the molecule is a ligand for an E3 ligase called MDM2. The two ends were linked by a PEG-based linker [12]. That small molecule PROTAC has enhanced cell penetration, marking the birth of the second-generation of PROTAC. Over the next few years, a series of PROTACs based on different E3 ligases sprang up (Figure 1). For example, due to the discovery that immunomodulatory drugs (IMiDs), including thalidomide, lenalidomide and pomalidomide, could act as ligands for the E3 ligase CRBN, a large number of CRBN-based PROTACs were successfully developed to degrade BRD4 [13], CDK9 [14], Sirt2 [15], KRAS^G12C^ [16], AKT [17], HDAC6 [18] and CDK6 [19]. Similarly, VHL-based PROTACs have successfully degraded ERRα [20], RIPK2 [21], HaloTag7 fusion proteins [22], BRD4 [23], Smad3 [24] and RTK [25], and RNF114-based PROTACs have degraded BRD4 [26] and BCR-ABL [26,27]; IAP-based PROTACs have degraded RIPK2 [28], CDK4/6 [29], Bcl-XL [29], etc. In more recent years, the explosive growth of PROTACs has indicated that PROTAC is becoming the most popular drug development technology. During the course of its development, PROTAC has constantly rectified some of its shortcomings and evolved into several novel targeted degradation technologies, which are summarized in the following (Table 1).

## 3. TF-PROTACs

Transcription factors (TFs) are a class of proteins that can specifically recognize and bind to the upstream of the gene to regulate the expression of the gene. TFs are closely related to cell proliferation and tumorigenesis, and, many TFs are considered as potential targets for tumor therapy [30]. However, unlike kinases or other enzymes, TFs usually do not have active sites or allosteric regulatory pockets, so traditional therapies using small molecule inhibitors targeting TFs have been unavailable for decades [31]. Delightfully, numerous studies have identified and predicted the unique DNA binding motif of most TFs, and sequences of DNA specific binding to TFs have also been explicated. Therefore, the team of Wenyi Wei utilized these characteristics to degrade TFs selectively and termed this platform as TF-PROTACs [32]. They linked oligonucleotides to the ligand of E3 ligase VHL via small molecule by chemical reactions, establishing NF-κB-PROTAC and E2F-PROTAC, successfully reducing endocellular p65 and E2F1, respectively (Figure 2). 

## 4. Light-Controllable PROTAC

PROTAcs have a promising prospect in clinical application due to their powerful targeted degradation function; however, its potential risk is the on-target off-tissue and off-target effect. Once PROTACs take effect in normal tissue, they may become unsafe, accompanied by toxic side effects. To solve this problem, many studies are trying to make PROTACs spatiotemporally controllable.

Photocaged PROTACs, including opto-PROTAC and pc-PROTAC, were reported first. Photocage is a group of molecular species that can be photodegraded, so after being connected to PROTAC, PROTAC activity can be controlled by light. Photocage group acts like a cage that inactivates the PROTAC, and when light is illuminated, the group is photodegraded, releasing active PROTAC. For instance, opto-PROTAC was invented in 2019 by Jian Jin and Wenyi Wei’s team [33]. In their research, since pomalidomide can bind to the E3 ligase CRBN in cells, researchers added a photolabile caging group on pomalidomide to block the binding. Conversely, the molecular group can be degraded by ultraviolet A (UVA), releasing free and active pomalidomide. In addition, they also modified dBET1 and ALK inhibitors to become photo-controlled PROTACs and called them Opto-DBET1 and Opto-DALK. Activity experiments showed lower toxic side effects and more controllable photocatalytic activity. In summary, light-controlled opto-PROTAC is likely to be more suitable than PROTAC in precision medicine. In parallel, derived from the known BRD2-4 degrader dBET1, Xue et al. developed photoactivatable PROTACs, termed photo-caged PROTACs (pc-PROTACs) [34]. They reduced the binding between dBET and BRD4 by introducing BBP to dBET1. Although inactive in the dark, the compound efficiently degraded BRD4 in Ramos cells after 3 min of 365 nm irradiation. They also applied this compound to zebrafish and successfully induced BRD4 degradation during embryogenesis.

Although photocage PROTACs can enable induced degradation, there is still a disadvantage; that is, the induced effect is irreversible, and once activated, its inactivation cannot be controlled. Several optical-switch PROTACs, including PHOTACs, AZO-ProTACs and photoPROTACs, have provided a remedy to this shortcoming. They designed the groups on PROTACs to enable the PROTAC to photochemically isomerize under the irradiation of light at different wavelengths, so that PROTAC can be reversibly active or inactive. For instance, photoswitchable PROTACs (photoPROTACs) by including ortho-F4-azobenzene linkers between POI ligand and E3 ligase ligand have been reported by Pfaff et al. [35]. Reynders et al. also developed PHOTACs to target either BET family proteins (BRD2, 3, 4) or FKBP12 [36]. In addition, Jin et al. invited Azo-PROTACs to degrade BCR-ABL fusion and ABL proteins in myelogenous leukemia K562 cells [37]. These studies have been successful in making protein degradation reversible.

## 5. PhosphoTAC

Phosphorylation is one of the most important post-translational modifications of proteins. Abnormal phosphorylation of some key proteins in cells is closely related to the occurrence of various diseases such as cancer and Alzheimer’s disease [38,39]. Phosphorylation is a reversible dynamic process regulated by both kinases and phosphatases. The former can catalyze the phosphorylation of proteins, whereas the latter catalyzes the dephosphorylation of proteins. In the early stages, people’s efforts are aimed at developing a series of small molecule inhibitors for protein kinases/phosphatases, but that strategy has obvious deficiencies. First, inhibitors of many kinases are difficult to design. Second, because inhibitors usually target the ATP binding sites of kinases, and it can also interact with the ATP binding sites of other kinases, off-target toxicity is prone to occur. Third, the mutation of the kinase binding site can easily lead to drug resistance. Thus, treatment strategies for abnormal protein phosphorylation need to be improved further.

Based on the basic principles of PROTAC, the Crews research group designed phospho-dependent proteolysis targeting chimeras (phosphoPROTACs) [40]. The characteristic of phosphoTAC is that the activity depends on whether the receptor tyrosine kinase (RTK) in the signal pathway is activated. Only phosphoTAC activated by the RTK can degrade the targeted protein. Specifically, TrkA and ERBB2/ERBB3 are two different types of RTK in cells. After being activated, both TrKA and ERBB2/ERBB3 can form dimers and autophosphorylates at tyrosine residues. The tyrosine phosphorylation sequences of TRKA and ERBB3 proteins are coupled to ligands of E3 ubiquitin ligase Von Hippel Lindau (VHL) via an aminohexanoic acid linker, forming phosphoTAC. When phosphoTAC is phosphorylated by activated RTK, the two phosphoPROTACs become active and recruit factor receptor substrate 2α (FRS2α) and phosphatidylinositol-3 kinase (PI3K), respectively, leading them to ubiquitination degradation. Both the two phosphoPROTACs demonstrate their ability to inhibit the signal transmission of their respective receptor tyrosine kinase pathways in vitro and in vivo. In addition, phosphoTAC is deactivated after mutating tyrosine to phenylalanine, demonstrating that activation of phosphoTAC is entirely dependent on kinase-mediated phosphorylation. Moreover, stimulation of unrelated growth factor receptors could not induce targeted protein degradation, indicating that phosphoTAC is specific in signaling pathways.

## 6. PhosTAC

In 2021, Crews introduced the concept of phosphorylation targeting chimeras (PhosTACs) [41], which is a more precise method of manipulating the activity of targeted proteins by dephosphorylation of targeted proteins rather than degrading them. PhosTAC and PROTAC are similar mechanisms, but the difference is that phosTAC recruits phosphatase instead of E3 ligase. Therefore, PhosTAC does not degrade the targeted protein but dephosphorylates the phospho substrate to regulate the protein’s function. This is a more precise means of controlling the activity of the targeted protein. In that study, phosphatase was fused with FKBP12 (F36V) and substrate protein was fused with Halo Tag. Meanwhile, they linked FKBP12 (F36V) ligand to the Halo Tag (see the previous one) ligand (chloroalkane) via linkers of poly (ethylene glycol) (PEG) groups such as phosTAC complex. They demonstrate that the corresponding PhosTAC complex successfully mediates the dephosphorylation of substrates such as PDCD4 and FOXO3a. Unlike protein degradation induced by traditional PROTAC, PhosTAC manipulates specific protein functions in a more precise manner, providing more options for protein processing (Figure 3).

## 7. LYTAC

Although PROTAC has many advantages, its shortcomings cannot be ignored. For example, PROTAC targets intracellular proteins, leaving secreted and cell membrane proteins unregulated. In 2020, this limitation was broken up by Professor Carolyn R. Bertozzi, who designed and validated the LYTAC technique using lysosomal degradation pathways to degrade secreted and cell membrane proteins [42]. There are lysosome-targeting receptors (LTRs) on the cell membrane, which can mediate protein transport to lysosomes for subsequent degradation [43]. The cation-independent mannose-6-phosphate receptor (CI-M6PR) is a typical LTR, which can transport proteins bearing N-glycans capped with mannose-6-phosphate (M6P) residues to lysosomes for degradation [44]. Based on this, they designed the LYTAC system for targeted degradation of secreted proteins and membrane proteins. The LYTAC system consists of two main parts; one is the glycopolypeptides containing M6P branches that can bind to the LTR, and the other is the specific antibody corresponding to the substrate protein to be degraded. They optimized the LYTAC system and successfully degraded EGFR, PD-L1 and CD71. Their studies demonstrated a novel strategy to degrade secreted and cell membrane proteins, greatly expanding the range of protein degradation.

## 8. AUTAC

Arimot et al. designed an autophagy-targeting chimera (AUTAC) [45], which mainly consists of two parts: one is the S-guaylation tag, and another is the small molecule ligand of the target protein. The two parts are connected through a polyethylene glycol (PEG) linker. The S-guanylation tag could induce polyubiquitination of the substrate, which can act as a degradation “label” to recruit autophagosomes for degradation; and the small molecule ligands can specifically recognize the target protein. They designed a series of AUTACs to specifically induce degradation of some proteins in cells, including the cytosolic protein methionine aminopeptidase 2 (MetAP2), FK506-binding protein (FKBP12) and the nuclear protein Brd4, and they also induced mitochondrial autophagy by the outer mitochondrial membrane protein Translocator protein (TSPO). However, the process of autophagy is much more complicated than PROTAC, and there are many influencing factors, so further improvement is needed.

## 9. ATTEC

In 2019, Professor Lu’s team developed an autophagosome-tethering compound (ATTEC) to degrade targeted proteins [46]. Similar to AUTAC, ATTEC is based on an autophagy-lysosome system. But the difference is that one end of ATTEC targets a specific protein, such as Mutant huntingtin protein (mHTT), a pathogenic protein of Huntington’s Disease. The other end directly targets microtubule-associated protein 1A/1B light chain 3 (LC3), a specific protein on the surface of the autophagy. The compound can connect the pathogenic targeted protein with LC3, then wrap the pathogenic protein into the autophagosome, and finally transport it to the lysosome to degrade the targeted protein.

In 2021, Professor Lu’s team developed a new ATTEC [47], named LD-ATTEC, which specifically degrades lipid droplets, an organelle that stores lipid in cells. LD-ATTEC utilized the previously reported LC3-binding molecule GW/DP as the ligand for targeting autophagosomes, and Sudan III/Sudan IV as the ligand for targeting lipid droplets, connecting the two ligands with a short linker. LD-ATTEC can specifically transport lipid droplets to autophagosomes, where the transported lipid droplets are degraded by lysosomes. This research achieved targeted degradation of lipid for the first time, suggesting that ATTEC can degrade non-protein macromolecules, providing a new approach for treating obesity-related metabolic diseases.

## 10. CMA

CMA is a kind of autophagy that can mediate protein degradation in cells. The main mechanism of CMA is that there is a KFERQ pentapeptide motif on the substrate protein, which can be specifically recognized and combined with the chaperone heat shock protein HSC70 (heat shock cognate protein of 70KDa) in the cytoplasm to form a complex. Then, the complex binds to lysosomal-associated membrane protein 2A (LAMP2A), resulting in the multimerization of monomeric LAMP2A. Finally, polymerized LAMP2A allows the complex to enter the lysosomal lumen for protein degradation [48].

Based on the mechanism of CMA, Wang’s group designed a novel chimeric peptide method and successfully degraded targeted proteins in cells [49]. The chimeric peptide consists of three domains, including a target protein binding domain (PBD), a CMA-targeting motif (CTM) and a cell membrane penetrating domain (CMPD). PBD is a peptide that can specifically bind to the POI, CTM is a KFERQ-containing motif, which can guide the POI to HSC70 and CMPDs can assist the chimeric peptide to penetrate the cell membrane and blood–brain barrier. These three domains enable the chimeric peptide to be efficiently and specifically recognized by HSC70 and degraded by the CMA pathway. The chimeric peptides were shown to be able to degrade death-associated protein kinase 1 (DAPK1), α-synuclein and Post Synaptic Density Protein 95 (PSD-95), which is a new strategy for endogenous protein degradation.

## 11. RNA-PROTAC

In 2019, RNA-PROTAC was developed to degrade RNA binding proteins (RBPs) [50]. RBPs are a class of proteins in cells that can bind to specific RNA, and abnormalities of RBP are associated with a variety of diseases [51]. Because RBP comprises domains that can bind to specific oligonucleotide sequences, this research makes use of the oligonucleotide sequence as POI ligands. Taking the stem cell factor Lin28 as an example, there is a zinc finger domain of its C-terminal, which can bind to microRNAs containing the 5’-AGGAGAU-3’ sequence. The sequence was employed as the ligand of Lin28. The peptide derived from the HIF-1-α transcription factor was selected as a ligand for recruiting E3 ligase VHL. After the combination and optimization of the two ligands, RNA-PROTAC can competitively bind to LIN28 with 5 ‘-AGGAGAU-3’-containing microRNAs, and recruit VHL for UPS-dependent ubiquitination degradation of Lin28. 

## 12. RIBOTACs

In 2018, Disney et al. created the ribonuclease targeting chimera (RIBOTAC). This small molecule can specifically recruit the endogenous ribonuclease RNase L to a specific RNA target, and then successfully eliminate the RNA by utilizing the intracellular nucleic acid cleavage mechanism [52]. Recently, the team of Disney updated the research, in which they transformed Dovitinib, a receptor tyrosine kinase (RTK) inhibitor, into a small-molecule RIBOTACs to degrade pathogenic RNA with a 2500-fold increased selectivity and reduced toxicity [53]. Subsequently, it has been demonstrated to have a significant anti-tumor effect in animal experiments.

## 13. Clinical Progress 

In recent years, with the successful degradation of previously considered “undruggable” pathogenic proteins by PROTAC, people have begun to apply PROTACs in clinics for the treatment of diseases, such as tumors, autoimmune disease, neurodegenerative diseases, alopecia, acne and asthma. There are also studies using this technology for treatment of Alzheimer’s disease and COVID-19/severe acute respiratory syndrome worldwide [54,55,56]. Thousands of PROTACs have been developed to degrade kinases, nuclear receptors, transcription factors, regulatory proteins and so on, among which the most popular targets include BET, AR, BTK, ALK, ER, MEK, Bcr-abl and EGRR. Table 1 has summarized the PROTACs entering the clinical trials stage. Benefitting from the progress of high-throughput screening (HTS), virtual screening, structure-based drug design (SBDD) and fragment-based drug design (FBDD), PROTAC will very likely evolve more rapidly and completely. In 2021, a team proposed a systematic approach to assessing the PROTAC tractability (PROTACtability) of protein targets [57], proposing “the PROTACtable genome”. They evaluated 19498 proteins and identified 1067 potential PROTAC targets that have not been reported in the literature but may have the opportunity to be PROTAC targets, providing powerful guidance for PROTAC-based drug development in the future (Table 2).

## 14. Current Challenges and Future Prospects of Degradation Technologies Based on PROTAC Mechanism

### 14.1. More Diversity

Up to now, the E3 ligase used by PROTAC are mainly CRBN, VHL, IAP and MDM2. There are also a few using DCF15, RNF114, DCAF16 and FEM1B. However, compared to the more than 600 E3 ligases predicted in vivo, only a few E3 ligases are in PROTAC platform. In addition to developing more E3 ligases, more suitable target protein ligands and linkers are also necessary for PROTAC, which requires accurate analysis and prediction of the structures of these molecules. Excitingly, in 2021, a breakthrough occurred in the field of protein structure prediction: AlphaFold and RoseTTA-fold, two AI-powered protein prediction technologies based on deep learning, which can quickly and accurately predict various complex protein structures based on limited information. The breakthrough is unprecedented in both speed and accuracy, and is bound to accelerate the development of more PROTACs greatly.

Although advances in technology have made the design of PROTAC much more convenient, finding a POI ligand is still not easy. In addition, even if applicable, if ligands are found, considerable efforts are still required to test various combinations of the linker and suitable E3 ligases. Finally, synthetic PROTAC molecules are limited by cellular permeability. Partridge’s team innovatively solved this problem by designing the bioPROTAC. Unlike traditional small-molecule PROTAC, bioPROTAC uses genetic engineering technology to express PROTAC-effect proteins in cells to drive POI degradation directly. Further, because bioPROTAC is an artificially engineered gene-encoded product, it can fuse POI ligands to any of the more than 600 E3 ligase enzymes, making full use of the cell’s UPS system. When the gradually improved mRNA/proteins delivery technology in vivo reaches maturity, bioPROTAC will probably be the easiest and most powerful PDT drug to design in the future.

In addition to bioPROTAC, molecular glues with the simpler chemical structures are excellent TPD molecules. Molecular glues degrade POI by inducing or stabilizing (enhancing) protein–protein interactions between E3 ubiquitin ligases and POI. The advantage of molecular glues is less spatial interference. It is believed that more diversified PDT will emerge in the future.

### 14.2. Suitable for More Diseases

As mentioned above, PROTAC has been used in the treatment of tumors, autoimmune diseases, neurodegenerative diseases, hair loss, acne, asthma and SARS-COVID-19. At present, there is another serious disease that plagues human beings: the infection of drug-resistant bacteria caused by the abuse of antibiotics. It is estimated that by 2050 the number of deaths will increase to 10 million per year [58]. This severe situation may be broken by PROTAC in the future. For instance, hijacking the protein quality control systems of drug-resistant bacteria and fungi can be a way to degrade their critical proteins and eliminate infections. Similarly, PROTAC may be a potent antiviral strategy in the future, such as blocking viral replication by degrading proteins critical to the process, which could lead to new treatments for diseases caused by viruses, including cervical cancer and oropharyngeal cancer caused by human papillomavirus (HPV), liver cancer caused by hepatitis C virus (HCV), Burkitt’s lymphoma caused by Epstein-Barr virus, Kaposi’s sarcoma caused by human herpesvirus 8 (HH8), adult T-cell leukemia caused by human T-lymphocyte virus (HTLV) and skin cancer (Merkel cell carcinoma) caused by Merkel cell polyoma virus. In the future, PROTAC is expected to become a new treatment for diseases caused by invading microorganisms, including drug-resistant bacteria, fungi and viruses.

### 14.3. Stronger Druggability

Insufficient solubility and cell penetration are the main obstacles to the PROTAC drugs. In addition, there are still many unknowns when PROTAC is used as a drug in vivo. For example, does the structure/activity of the theoretically “recyclable” PROTAC molecule change after it has performed its function in the cell? If yes, what are the differences? Are there any consequences as the result of the changes? In addition, although a variety of controllable “PROTACs” has emerged, using UVA as a controller is very likely to damage DNA and is difficult to treat deep lesions in the body. How to make PROTAC more precisely controllable in both cells and tissues? These are all challenges that need to be faced. In the future, a more efficient and convenient mode of drug delivery with higher safety, efficacy and selectivity is a development direction of PROATAC.

### 14.4. Based on More Mechanisms 

Similar to the existing hijacking ubiquitin-proteasomes, lysosomes, RNAse, molecular chaperones and phosphatases in cells to target and regulate proteins, some other regulatory mechanisms existing in cells can also be theoretically utilized, e.g., hijackingintracellular glycosylation, phosphorylation, methylation, deglycosylation, dephosphorylation and other related enzymes to target and regulate the post-translational modification of POI. Other possibilities can be hijacking the cellular allosteric proteins to target and regulate conformation of POI, or, the proteins related to protein spatial localization and transport in cells to regulate the subcellular localization of POI. In conclusion, an observable feasibility is there in making use of these mechanisms to precisely regulate the activity and function of POI in the future.

## 15. Conclusions

Since PROTAC was first designed and synthesized for protein degradation in 2001, more and more degradation technologies based on PROTAC mechanism have emerged, leading to new waves of drug development. Although PROTACs still have many problems to be solved, it is believed that in the future, with the development of technology and the deepening of research, the design and synthesis of degradation technologies will be gradually optimized, and eventually open up a broad road for the treatment of various diseases.

## Figures and Tables

**Figure 1 biomolecules-12-01257-f001:**
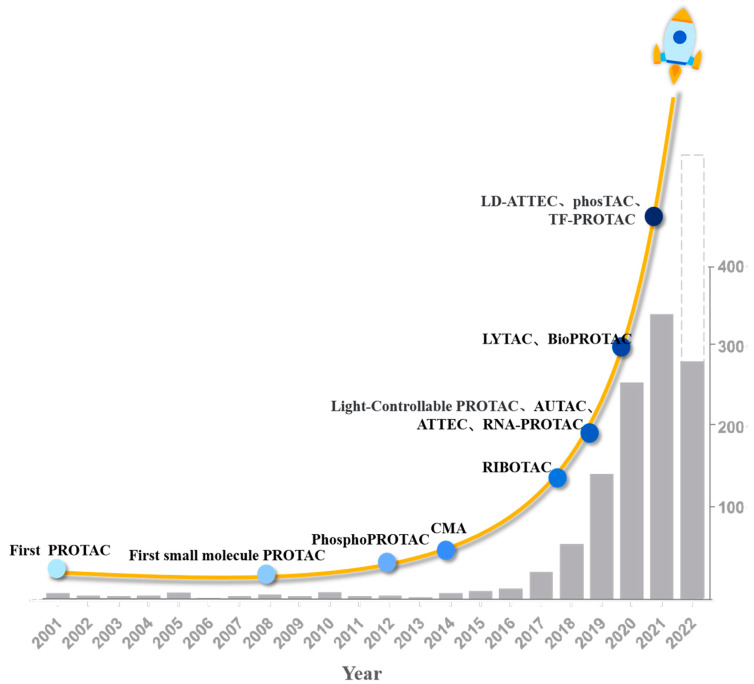
Timeline of milestones in protein degradation technologies (PDT) based on PROTAC development (2001–2022) and rough statistics on the number of PROTAC publications in PubMed (accessed in August 2022). The abscissa is the year and the ordinate is the quantity of publications of PROTACs. Since the first PROTAC was reported in 2001, both species and quantity of PDT have been greatly developed, especially in recent years. Among the more than 900 papers on PROTAC published at present, there are 786 papers in the three years from 2019 to 2021, accounting for about 80%, which is four times the number of papers published from 2001 to 2018. As of August 2022, 287 articles have been published, and the possible number of publications in this year was estimated by dotted lines.

**Figure 2 biomolecules-12-01257-f002:**
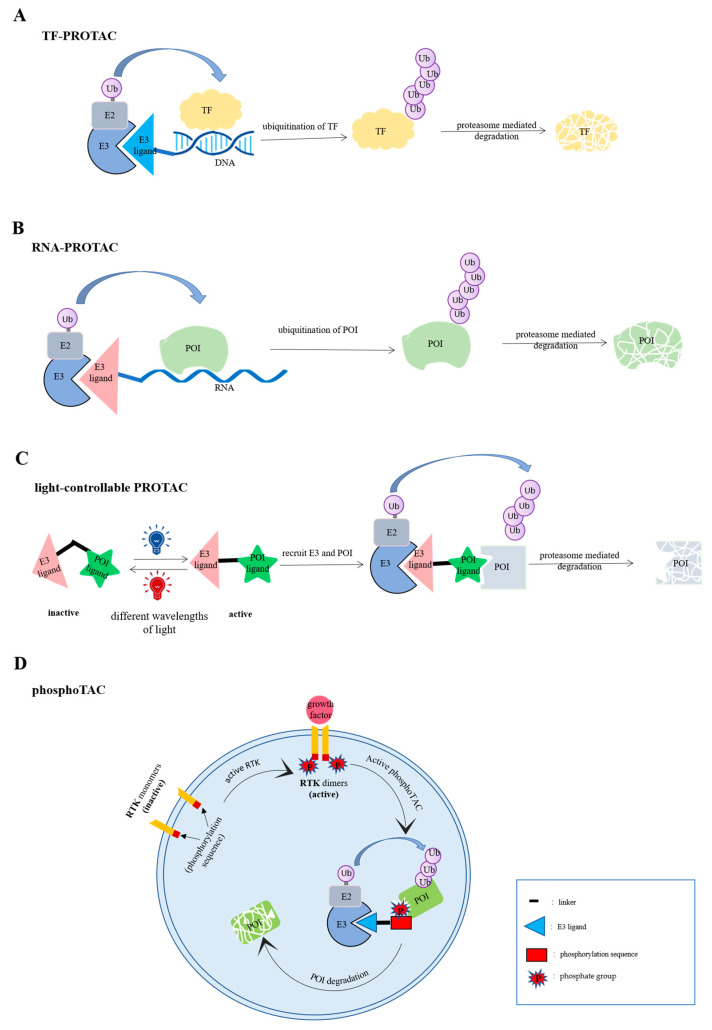
Schematic diagram of targeted degradation strategies based on ubiquitin-proteasome system (UPS). (**A**) The DNA sequence of TF-PROTAC selectively binds to transcription factor (TF), whereas E3 ligase ligand specifically recruits E3 ligase. Finally, TF is ubiquitinated by E3 ligase and degraded by the proteasome. (**B**) RNA-PROTAC binds to POI via RNA sequence, and recruits E3 ligase specifically by E3 ligase ligand. After being modified by ubiquitination, POI is degraded by the proteasome. (**C**) Light-controllable PROTAC photochemically isomerize upon irradiation with different wavelengths of light. Therefore, the PROTAC can be reversibly active or inactive. Only when PROTAC is activated by a specific wavelength of light, it can degrade POI. (**D**) RTK monomers are inactive. When RTK is activated by growth factor, in addition to dimerization and autophosphorylation, RTK can also induce PhosphoTAC activation, which enables phosphoPROTACs to recruit POI and E3 ligase, and ultimately leads to ubiquitination and degradation of POI.

**Figure 3 biomolecules-12-01257-f003:**
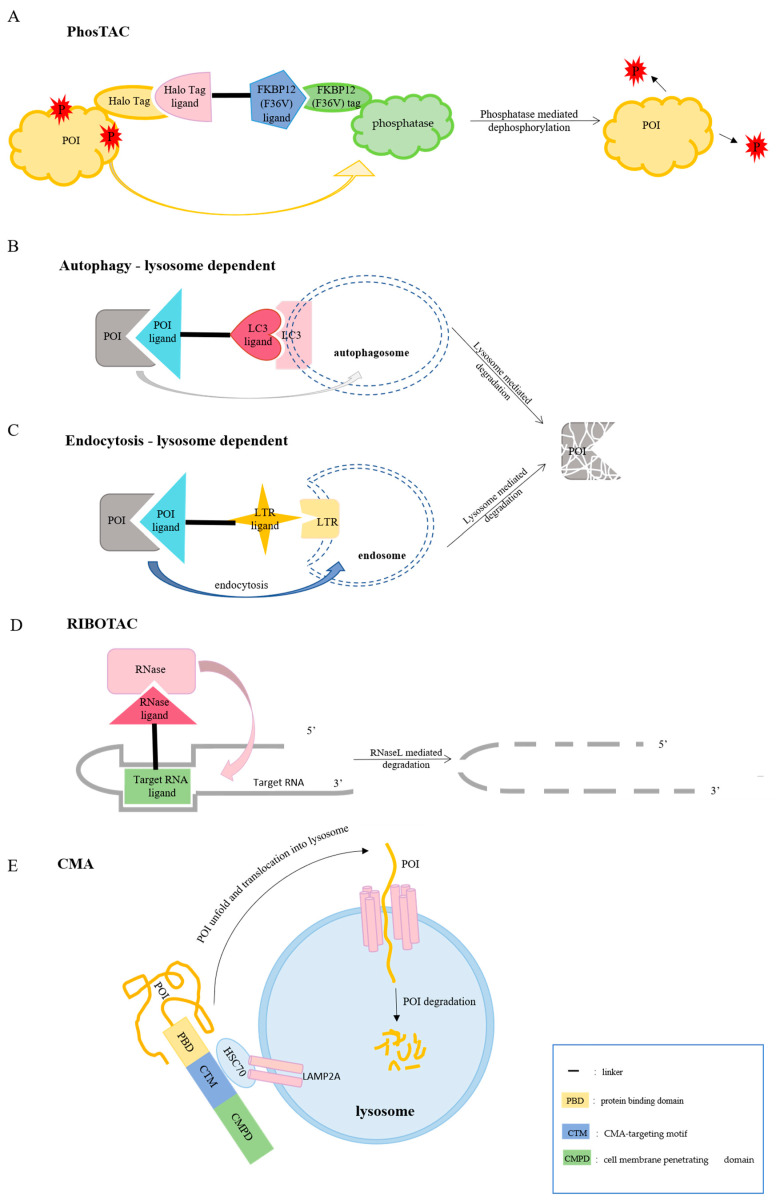
Schematic diagram of targeted degradation strategies based on other mechanisms. (**A**) PhosTAC induces dephosphorylation of POI. The mechanism is that the Halo Tag ligand of PhosTAC can specifically bind to the POI fused with HaloTag, and another FKBP12 (F36V) ligand can recruit the phosphatase fused with FKBP12 (F36V) tag. Ultimately, phosphatases catalyze the dephosphorylation of POI. (**B**) Autophagy–lysosome-dependent PROTACs connect the POI with LC3, a specific protein on the surface of the autophagy, then wrap the POI into the autophagosome, and finally transport it to the lysosome for degradation. (**C**) LYTACs can specifically bind to extracellular/membrane POI, and bind to the lysosomal targeting receptor LTR of the cell membrane, so that POI enters the endosome through endocytosis and finally enters the lysosome for degradation. (**D**) RIBOTACs specifically recruit the endogenous ribonuclease RNase L to a specific RNA target, and then successfully eliminate the RNA by RNaseL. (**E**) The CMA-based chimeric peptide contains protein binding domain (PBD) and CMA-targeting motif (CTM), which can respectively bind to POI and HSC70. In addition, the chimeric peptide also contains a domain that facilitates membrane penetration, named CMPD. When HSC70 interacts with LAMP2A on the surface of the lysosomal membrane, the POI unfolds and enters the lysosome, where it is finally degraded.

**Table 1 biomolecules-12-01257-t001:** The advantages and disadvantages of each technology.

Technology	Degradation Mechanism	Advantage	Disadvantage
TF-PROTACs	ubiquitin-proteasome system	TFs without active sites or allosteric regulatory pockets can be degraded by TF-PROTAC.	It is difficult to design TF-PROTAC with an unknown DNA-binding sequence.
Light-controllable PROTAC		The activity can be controlled bydifferent lights.	The PROTAC may be not effective for deep tissue that light cannot penetrate.
PhosphoTAC		The activity of PROTAC is dependent on the phosphorylation of the signal pathway.	Mutations of phosphorylation sites may affect the activity of PhosphoTAC.
RNA-PROTAC		RNA-PROTAC specifically degrades RNA-binding proteins.	RNA-PROTAC may be easily degraded since RNA is unstable.
PhosTACs	phosphatase	Compared with degrading the target protein, PhosTAC induces the dephosphorylation of the target protein, which is a more precise way of regulating the protein.	Protein dephosphorylation induced by PhosTAC is only applicable to diseases caused by abnormal phosphorylation.
CMA	CMA-lysosome	The peptide of CMA is easy to design.	CMA is chimeric polypeptides, so it has poor transmembrane ability and low stability.
LYTAC	Endocytosis-lysosome	LYTAC can induce targeted degradation of secreted and cell membrane proteins	LYTAC is not stable enough in vivo. In addition, the LTR ligands of LYTACs are chemically synthesized sugar, which may produce strong immunogenicity in the body.
AUTAC	autophagy-lysosome	induce the degradation of proteins and organelles by lysosomes.	The degradation process is complicated and there are many influencing factors.
ATTEC		ATTEC can degrade not only proteins but also lipid droplets. In addition, ATTEC molecules are small, so it is easy to penetrate cell membranes.	Whether the ATTEC will affect the overall autophagy activity and how to avoid the non-specific degradation of autophagy-related proteins remains to be further explored.
RIBOTAC	RNaseL	RIBOTAC selectively degrades target RNA.	It is difficult to develop target RNA ligands.

**Table 2 biomolecules-12-01257-t002:** The PROTACs entering the clinical trials.

Drug	Company	Targeted Protein	Indication	Stage of Clinical Trial
ARV-110	Arvinas	Androgen receptor (AR)	Metastatic castrate resistant prostate cancer	Phase II
ARV-471	Arvinas	Estrogen Receptor-α(ER-α)	ER+/HER2-Breast cancer	Phase II
ARV-766	Arvinas	Androgen receptor (AR)	Metastatic castrate resistant prostate cancer	Phase I
DT2216	Dialectic	BCL-XL	Liquid and solid tumors	Phase I
KT-474	Kymera/Sanofi	Interleukin 1 receptor associated kinase4(IRAK4)	Autoimmune diseases	Phase I
NX-2127	Nurix	Bruton tyrosine kinase (BTK)	B-cell Malignancies, including CLL, SLL, WM, MCL, MZL, FL, DLBCL	Phase I
NX-5948	Nurix	Bruton tyrosine kinase (BTK)	B-cell Malignancies, including CLL, SLL, DLBCL, FL, MCL, MZL, WM, PCNSL	Phase I
FHD-609	Foghorn	Bromodomain containing9(BRD9)	Synovial Sarcoma	Phase I
HSK29116	Haisco	Bruton tyrosine kinase (BTK)	B-cell Malignancies	Phase I
BGB-16673	BeiGene	Bruton tyrosine kinase (BTK)	B-cell Malignancies	Phase I
AR-LDD	Bristol Myers Squibb	Androgen receptor (AR)	Prostate Cancer	Phase I
KT-413	Kymera	Interleukin 1 receptor associated kinase4(IRAK4)	MYD88-mutant Diffuse Large B-Cell Lymphoma	Phase I
KT333	Kymera	signal transducer and activator of transcription 3	Liquid and solid tumors	Phase I
GT-00029	Kintor	Androgen receptor (AR)	Androgenetic alopecia and acne	Phase I
AC0682	Accutar	Estrogen Receptor (ER)	Breast cancer	Phase I
AC0176	Accutar	Androgen receptor (AR)	Prostate cancer	Phase I

## Data Availability

Not applicable.

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
