# Peer review of "Recent Advances of Degradation Technologies Based on PROTAC Mechanism"

_biomolecules, 2022, doi:10.3390/biom12091257_

Round 1
Reviewer 1 Report
This brief, poorly illustrated review reads like a list of different protein degradation technologies, which adds very little to a vast body of reviews on PROTACs and other protein degradation technologies. Many of the advantages of PROTacs are not discussed.
Author Response
Question: This brief, poorly illustrated review reads like a list of different protein degradation technologies, which adds very little to a vast body of reviews on PROTACs and other protein degradation technologies. Many of the advantages of PROTACs are not discussed.
Response: We are grateful for your comments on the manuscript. According to your advices, we have added the advantages of PROTACs to the second paragraph of the manuscript on page
- This review starts from the birth of PROTAC, and describes the advances of novel types of degradation technologies based on PROTAC mechanism. Last but not the least, the novelty of this review is the systematic comparison of the advantages and disadvantages of various degradation technologies, in addition to putting forward many innovative and interesting ideas for the future development of RPOTAC. Thanks again for your comments and we look forward to hearing from you!

Reviewer 2 Report
The authors presented a review paper on the Novel Types of Protein Degradation Technologies
The abstract is to be extended.
As review paper, the manuscript is relatively short.
The novelty of the present work is to be stated.
What is the sub-title of the first paragraph of the paper ? is it ‘’Introduction’’?
A bibliometric study is to be added.
The paper is to be checked against misprints and grammatical mistakes.
A list of abbreviations is to be added.
Is there any standard and regulation related to Protein Degradation Technologies?
Some figures presenting the used technologies in the reviewed papers are to be added.
A table summarising the advantages and disadvantages of each technology is to be added.
Conclusion is missing.
Author Response
Question1. The abstract is to be extended.
Response: We thank you very much for the suggestions. As suggested, We have extended the abstract in the revised version of the manuscript.
Question2.As review paper, the manuscript is relatively short.
Response: The new version of the manuscript has been added as requested with new contents, tables and figures.
Question3. The novelty of the present work is to be stated.
Response: We have stated the novelty of the present work in the introduction section. We start with PROTAC technology, the earliest and fastest developing protein degradation technology (PDT), then review the progress of degradation technologies based on PROTAC mechanism in the past 20 years. Last but not the least, the novelty of this review is the systematic comparison of the advantages and disadvantages of various degradation technologies, in addition to putting forward many innovative and interesting ideas for the future development of RPOTAC.
Question4. What is the sub-title of the first paragraph of the paper? Is it “Introduction”?
Response: Yes, we have added "Introduction" as the sub-title of the first paragraph.
Question5. A bibliometric study is to be added.
Response: We gratefully appreciate your valuable suggestion. We have added the bibliometric study to the statistics of the articles in the legend of Figure3 on page 11.
Question6. The paper is to be checked against misprints and grammatical mistakes.
Response: Thanks for suggestion. We have checked and corrected the mistakes in the manuscript.
Question7. A list of abbreviations is to be added.
Response: The list of abbreviations was attached to the table 3 of the revised manuscript.
Question8. Is there any standard and regulation related to Protein Degradation Technologies?
Response: This is an interesting question. An article entitled "Prey for the Proteasome: Targeted Protein Degradation—A Medicinal Chemist's Perspective" (https://doi.org/10.1002/anie.202004310) mentioned the point. As written in the review, TPD is still in its infancy thus far only scratching the surface of its full potential. But within the near future we will see TPD develop as one of the standard procedures to verify hits identified via genome editing screens. With more TPD approaches already allowing an in vivo application targeting validation will raise to a completely new level.
Question9. Some figures presenting the used technologies in the reviewed papers are to be added.
Response: Thank you for your advice. We have added TF-PROTAC, RNA-PROTAC, light-controllable PROTAC, phosphoTAC, phosTAC, and CMA technologies to the previous figure, making the new more comprehensively cover of the used techniques. Please see Figures 1 and 2 on pages 8-11 for details.
Question10. A table summarising the advantages and disadvantages of each technology is to be added.
Response: Thank you very much for your suggestions. As suggested, the advantages and disadvantages of each technology had been summarized in table1 on page 6-7.
Question11. Conclusion is missing.
Response: Thank you for your suggestions. We have added the conclusion on page 14.

Reviewer 3 Report
The authors briefly summarized various PROTAC related techniques and provided some insights. Overall the manuscript is easy to read and covers all the major developments in the field. While the manuscript lacks the depth of analyzing each technology, it would serve as an introductory article for readers. There are some issues of not citing proper papers or missing key papers. The authors need to be more thorough on literature citations. There are also many typos in the manuscript that require extensive editing.
1) In the introduction part, no references were cited. The topic has been extensively reviewed, the authors at the minimum should cite some previous review papers. 2) When talked about the 2001 work by Crews and Deshaies, no reference was cited. In the same paragraph, the authors discussed various targets that had been targeted by CRBN-based, VHL-based, or other E3-based PROTACs. There are many missed or wrongly cited references. The authors need to check each citation (and throughout the manuscript). 3) In the clinical progress session, the authors essentially just listed a table without saying anything else. Not clear why references 45 and 46 should be included in this part. 4) There are also a number of author names misspelled. For examples, Jin Jian should be Jian Jin (Session 2); Wei Wenyi should be Wenyi Wei (Session 2); Matthew should Disney (Session 10).
Author Response
Question1. In the introduction part, no references were cited. The topic has been extensively reviewed, the authors at the minimum should cite some previous review papers.
Response: Thank you very much for your suggestions. As suggested, We have added some previous papers in the revised version of the manuscript.
Question2. When talked about the 2001 work by Crews and Deshaies, no reference was cited. In the same paragraph, the authors discussed various targets that had been targeted by CRBN-based, VHL-based, or other E3-based PROTACs. There are many missed or wrongly cited references. The authors need to check each citation (and throughout the manuscript).
Response:Thank you so much for your careful check. We have cited the reference published by Crews and Deshaies in 2001. In addition, we have checked and added some literature in the manuscript.
Question3 In the clinical progress session, the authors essentially just listed a table without saying anything else. Not clear why references 45 and 46 should be included in this part.
Response:
Thanks for your comments. The reference 45 has been replaced by another paper entitled “Efficient Inhibition of SARS-CoV-2 Using Chimeric Antisense Oligonucleotides through RNase L Activation”. Because reference 46 assesses the PROTAC tractability (PROTACtability) of protein targets, which may provide powerful guidance for PROTAC-based drug development in the future, so it is cited here
Question4. There are also a number of author names misspelled. For examples, Jin Jian should be Jian Jin (Session 2); Wei Wenyi should be Wenyi Wei (Session 2); Matthew should Disney (Session 10).
Response: Thank you for pointing out mistakes of the manuscript. The misspellings have been checked and corrected.

Round 2
Reviewer 1 Report
Addition of figures and tables improve the review, but English needs improvement.
Reviewer 2 Report
Accepted